# Substantially Greater Carbon Emissions Estimated Based on Annual Land-Use Transition Data

**Jiaojiao Diao** [1,2] **, Jinxun Liu** [3] **, Zhiliang Zhu** [4] **, Mingshi Li** [1,2,*] **and Benjamin M. Sleeter** [5]

1   College of Forestry, Nanjing Forestry University, Nanjing 210037, China; diaojiaojiao@njfu.edu.cn
2   Co-Innovation Center for Sustainable Forestry in Southern China, Nanjing Forestry University, Nanjing 210037, China
3   U.S. Geological Survey, Menlo Park, CA 94025, USA; jxliu@usgs.gov
4   U.S. Geological Survey, Reston, VA 20192, USA; zzhu@usgs.gov
5   U.S. Geological Survey, Seattle, WA 98402, USA; bsleeter@usgs.gov
*   Correspondence: nfulms@njfu.edu.cn

**Abstract:** Quantifying land-use and land-cover change (LULCC) effects on carbon sources and sinks has been very challenging because of the availability and quality of LULCC data. As the largest estuary in the United States, Chesapeake Bay is a rapidly changing region and is affected by human activities. A new annual land-use and land-cover (LULC) data product developed by the U.S. Geological Survey Land Change Monitoring and Analysis Program (LCMAP) from 2001 to 2011 was analyzed for transitions between agricultural land, developed land, grassland, forest land and wetland. The Land Use and Carbon Scenario Simulator was used to simulate effects of LULCC and ecosystem disturbance in the south of the Chesapeake Bay Watershed (CBW) on carbon storage and fluxes, with carbon parameters derived from the Integrated Biosphere Simulator. We found that during the study period: (1) areas of forest land, disturbed land, agricultural land and wetland decreased by 90, 82, 57, and 65 km$^2$, respectively, but developed lands gained 293 km$^2$ (29 km$^2$ annually); (2) total ecosystem carbon stock in the CBW increased by 13 Tg C from 2001 to 2011, mainly due to carbon sequestration of the forest ecosystem; (3) carbon loss was primarily attributed to urbanization (0.224 Tg C·yr$^{-1}$) and agricultural expansion (0.046 Tg C·yr$^{-1}$); and (4) estimated carbon emissions and harvest wood products were greater when estimated with the annual LULC input. We conclude that a dense time series of LULCC, such as that of the LCMAP program, may provide a more accurate accounting of the effects of land use change on ecosystem carbon, which is critical to understanding long-term ecosystem carbon dynamics.

**Keywords:** land-use and land-cover change; carbon stock; carbon emission; LUCAS; LCMAP

## 1. Introduction

Carbon sources and sinks from land-use and land-cover change (LULCC) are a sizeable component of the global carbon budget [1]. However, estimation of this component is highly variable and uncertain when compared to estimates in other sectors [1,2]. One of the major causes of such uncertainty is the strong interannual variability in LULCC [3,4], such as forest thinning and recovery or seasonal wet-and-dry cycles of wetlands, which have strong greenhouse gas (GHG) emission implications [5].

Multi-date land-cover observation data are needed to estimate land-cover-type transitions. However, these land change data are not always consistent and easily acquired. Many studies of land change have been limited to certain types of land cover (such as forest land), which represent only part of a complete land-cover archive [6,7]. In addition, most researchers have studied a subset

of land-cover change processes (such as logging and fires) and did not include sufficient types of land-cover transitions, such as agricultural transfers, urbanization, and forest management.

The Landsat series is a well-known source used to produce LULCC map products at 30-m resolution, such as the U.S. Geological Survey (USGS) National Land Cover Dataset (NLCD) [8]. NLCD is an effective map product for general land-cover applications and has been widely used in many scientific and management applications. However, there is a possibility that some episodic LULCC events were missed between two observation dates [9]. Thus, to obtain continuous LULCC events, higher-quality data with additional land-cover types, land change variables, and most importantly, more frequent land change information, are urgently needed. New LULCC data produced by the USGS Land Change Monitoring, Assessment and Projection (LCMAP) program has been recently released (https://www.usgs.gov/land-resources/eros/lcmap). LCMAP uses a modified version of the Continuous Change Detection and Classification algorithm as the primary model, and is designed to produce continuous annual LULCC data at 30-m resolution, albeit in a relatively simplified classification scheme [10]. A total of nine land-cover classes are available in LCMAP: developed, agriculture, grassland/shrubland, forest, wetland, ice and snow, barren, water body, and disturbed/transition. With overall agreement ranging from 73% to 88% [11,12], LULC resulting from LCMAP can be summarized in annual steps.

All available Landsat images from 2001 to 2011 were used as input. Clouds were masked by using the Fmask algorithm and multi-temporal analysis of Landsat data [13,14]. Land change detection is based on the Continuous Change Detection and Classification (CCDC) algorithm, which uses a threshold derived from all Landsat bands and can detect changes at high temporal frequency. Moreover, LCMAP production is fully automated and can monitor many types of land-cover change once new Landsat images become available [15]. Because of a much longer temporal history and greater temporal resolution than NLCD, LCMAP offers long-term, consistent landscape conditions. Moreover, CCDC improved the ability to characterize gradual land-surface changes for LCMAP, such as year-to-year forest thinning and recovery or seasonal wet-and-dry cycles of wetlands [11]. To simulate the carbon cycle, the accuracies of the land-cover classification and LULCC datasets are critical. Thus, the LULC of LCMAP was used to study land-cover and its changes, as well as their effects on carbon cycles, in this study.

Currently, for example, in the Global Carbon Program (GCP), predictive models that simulate the structure of vegetation, growth and carbon cycles in given climate and undersea conditions are used as the main tool for estimating terrestrial carbon sinks [16]. Many models have been used to estimate their impact on the carbon cycle, including the DGVM (Dynamic Global Vegetation Model) carbon cycle model [17] and bookkeeping model [1,16]. Based on ecosystem-specific growth and decay equations, bookkeeping models track all carbon pools to simulate the annual carbon change of vegetation and soil in forest ecosystems [1]. All DGVMs represent the process of vegetation growth and death as well as the decomposition of dead organic matter related to the natural cycle, including the response of vegetation and soil carbon to increased atmospheric $CO_2$ concentrations and climate variability. The impacts of environmental change and land use on the carbon cycle in such studies are often combined. However, carbon flux related to LULC is one of the most uncertain quantities in anthropogenic global carbon budgets [18,19].

Unlike most landscape simulation models developed for specific regions and questions, the State and Transition Simulation Model (STSM) is a general, randomized, and spatially explicit approach for projecting landscape dynamics [20]. STSM projects changes in LULC to divide the landscape into a set of raster simulation cells and randomly forwards the state and age of each cell based on chronology, in response to probability transfers. Carbon storage and flux changes are integrated into the STSM by adding a carbon dynamics stock-flow (SF) model. This integrated model is the Land Use and Carbon Scenario Simulator (LUCAS), which can be used to address many challenges, including those in ecological research, forest management, water source management, wetland methane [21]. With

parameters of carbon flow between various pools calibrated by the Integrated Biosphere Simulator model (IBIS), LUCAS is used to project land change and calculate carbon storage and fluxes [22].

Chesapeake Bay includes several main land-cover classes such as forest land, agriculture land, developed area and wetland. Forests have the ability to protect clean air and water, provide habitat to wildlife, store carbon, control floods, and support the regional economy (https://www.chesapeakebay.net/state/tree_cover). Almost all the forest land in the region has undergone forest cutting at one time or another. Although many of the forests were cleared in the 18th and 19th centuries for farmland, timber and fuel were allowed to regrow. Forests that were cleared in recent decades for development are considered permanently lost. Between 1990 and 2005, there was about 14,770 ha of forest land lost annually in the Chesapeake Bay watershed. This rate fell in 2006 to an estimated 25,550 ha annually but is still unsustainable.

In a case study of the present work, we used LCMAP initial test products, LUCAS, and IBIS in a pilot study of the southern Chesapeake Bay Watershed (CBW). Our objectives were to: (1) depict LULCC and carbon history from 2001 to 2011; (2) qualify the effects of LULCC on carbon balance; and (3) examine the effect of the LULCC time interval on the data on carbon emissions.

## 2. Materials and Methods

### 2.1. Study Area

The Chesapeake Bay area covers more than 168,000 km$^2$, with more than 100 rivers and thousands of tributary streams [23]. This bay area is a habitat for a large number of saltwater and fresh fauna and flora, which provide valuable economic benefits and ecological services [24]. It is also home to about 15.7 million people, most of whom live in and around major urban areas such as Baltimore, Maryland, Washington DC, and Virginia [25]. Urban expansion around Chesapeake Bay has greatly altered both terrestrial and aquatic areas. Strenuous efforts have been made to reduce the rates of exploitation of various natural lands, especially wetland. The study area is located in south of the CBW, in the eastern United States (Figure 1). It covers an area of ~22,500 km$^2$, with altitudes ranging from sea level on the coastal plains to ~100 m on the Piedmont Plateau. Temperature typically varies from 34 to 88°F and annual precipitation from 1020 to 1270 mm, falling more or less evenly throughout the year. The main land-cover types are forest and agriculture. Historically, the study area has lost some forest vegetation because of agricultural expansion and urban sprawl.

### 2.2. LULCC Data Resource

The LCMAP products are derived from the U.S. Landsat Analysis Ready Data (ARD) tile format, defined by the Albers Equal Area Conic map projection referenced to the WGS84 datum. The annual primary land-cover database with nine classes for the U.S. Landsat ARD of Chesapeake Bay (h28v09) from 1987 to 2015 produced at the end of 2017 was available for testing. The LCMAP provides nationwide data of LULCC at 30-m resolution, with a 9-class legend. To align with the results of IBIS, we resampled all the input LULC data to 960-m resolution.

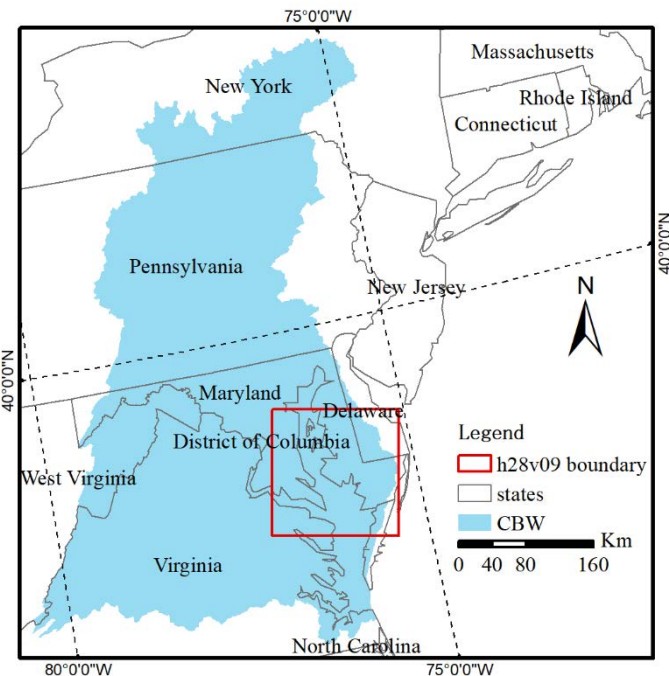

**Figure 1.** Location of study area (h28v09) in Chesapeake Bay Watershed (CBW).

*2.3. Research Procedure*

2.3.1. Procedure of Producing LULCC

The LULC products were converted to land-use change information. By comparing land-use type between two years using ArcGIS (v. 10.3), major land-use changes were identified. To study the LULC change and dataset for LULC conversion and carbon balance, we considered two scenarios: one without LULCC, and the other with LULCC resulting from an LCMAP annual land-cover dataset. Furthermore, to study the density effect of LULC on LULCC and carbon emissions, we considered and used three other scenarios: one of two to three years within the LULCs of 2001, 2004, 2006, 2008, and 2011; another of five years within the LULCs of 2001 and 2011; and one of 10 years within the LULCs of 2001 and 2011. All simulations were performed with version 3.2.13 of ST-Sim using SyncroSim software (version 2.2.4), with model inputs and outputs prepared by RStudio software version 1.1.463.

2.3.2. Carbon Cycle Modeling Approach

To illustrate the LUCAS method, we propose a model of CBW interaction between LULCC and terrestrial carbon. By adding a carbon dynamics inventory flow (SF) model, changes in carbon storage and flux were integrated into the framework of STSM. Then, the carbon model became an integrated model of LULCC with carbon reserves and emissions. The result was an integrated model of LULCC and carbon stock and emissions. The IBIS is a single, physically consistent modeling framework [26] that can be used to represent the main ecosystem processes that control vegetation and function, including plant physiology, land-surface physics, bio-geochemical cycling, canopy gas exchange, and vegetation competition [27]. It can determine detailed carbon pools and fluxes along with additional input data and parameter controls, which are very important for LUCAS to rely on and drive the exchange of carbon between pools [22].

2.3.3. Carbon Stock and Flows

We assumed that the annual carbon input from atmosphere to the ecosystem is the total net primary productivity (NPP) predicted by IBIS. Flux growth is the current type and age of coverage and is independent of any other carbon reserves in the model. The remaining flows are expressed as the

ratio of their respective source inventories (i.e., flow rates). With the exception of the discrete LULC class for each cell, each carbon pool is defined as a reserve type based on the LUCAS method (Figure 2). The carbon components included living biomass, standing deadwood, down deadwood, harvest wood production (HWP) extracted, straw, grain, new litter, litter, soil organic matter, and aquatic. The carbon pools of atmosphere and HWP extracted were included in the ecosystem to enforce mass balance. There are two pathway diagrams in the LUCAS model. One is transition between states (Figure 2a) (i.e., land-cover types) and the other carbon flows between carbon pools (Figure 2b). In addition, stock flow is a portion of the model, and flows are carbon fluxes between carbon stocks, including: (1) growth, mortality, harvest and litterfall of living biomass; (2) decay of deadwood; (3) decomposition and ageing of litter; and (4) emissions from living biomass, deadwood, raw litter, litter, and soil organic matter.

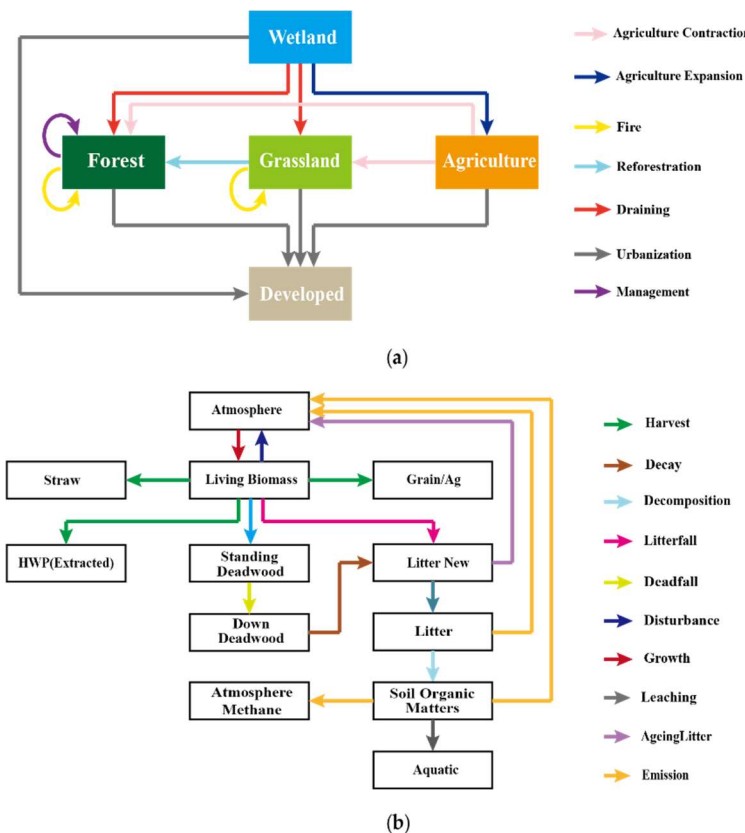

**Figure 2.** Flowchart showing carbon stocks (boxes) and flows (arrows) for terrestrial carbon budget model. Pathway diagrams for state-and-transition simulation model with stock and flows. (**a**) Transition diagram showing state types and transitions (arrows) for land-use/land-cover change model. (**b**) Flow diagram showing stocks (boxes) and flows (arrows) for terrestrial carbon budget model.

### 2.3.4. Model Initialization

The starting land cover of each simulation cell was derived from year 2001 of the LCMAP. We identified agricultural, forest, grassland, wetland and developed areas, and set the remaining land cells to "no-data". We derived the initial forest age map from forest age data of 2006, which was modified to represent the age of 2001 by subtracting 5 years from the forest age of 2006. Cells with a resulting forest age less than or equal to zero were input based on the maximum harvest rotation (50 years [28,29]) for the study area. The initial carbon of every carbon pool and carbon flux was derived from IBIS.

## 3. Results

### 3.1. Land-Cover Change in CBW

Given the effects of growing populations and human activities, the CBW has had large increases in urban sprawl. As shown in Figure 3c,d, some forest and agricultural pixels around the developed area in 2001 converted to developed pixels in 2011. Similarly, some forest pixels were transferred to agricultural land in 2011. Those transitions resulted in decreases of agriculture and forest and wetland areas but increases in developed land. In the study area, net agricultural land loss was about 57 km$^2$ during the study period from 2001 to 2011, with a loss of 180 km$^2$ and gain of 124 km$^2$ (Table 1). The major loss of agricultural land area resulted from conversion to developed area (68%) and forest land (17%). The area of wetland decreased from 1,728 to 1,663 km$^2$, which was mainly converted to forest (50 km$^2$) and developed (25 km$^2$) land, with 10 km$^2$ gained from forest. Forest land decreased from 6,441 to 6,351 km$^2$, of which 168 km$^2$ was converted to developed land. Within the CBW between 2001 and 2011, there was a 24.8% increase in developed land (from 1,316 to 1,609 km$^2$), most of which occurred in forest (57%) and agriculture (42%). Grasslands only increased by 1 km$^2$ over the study period.

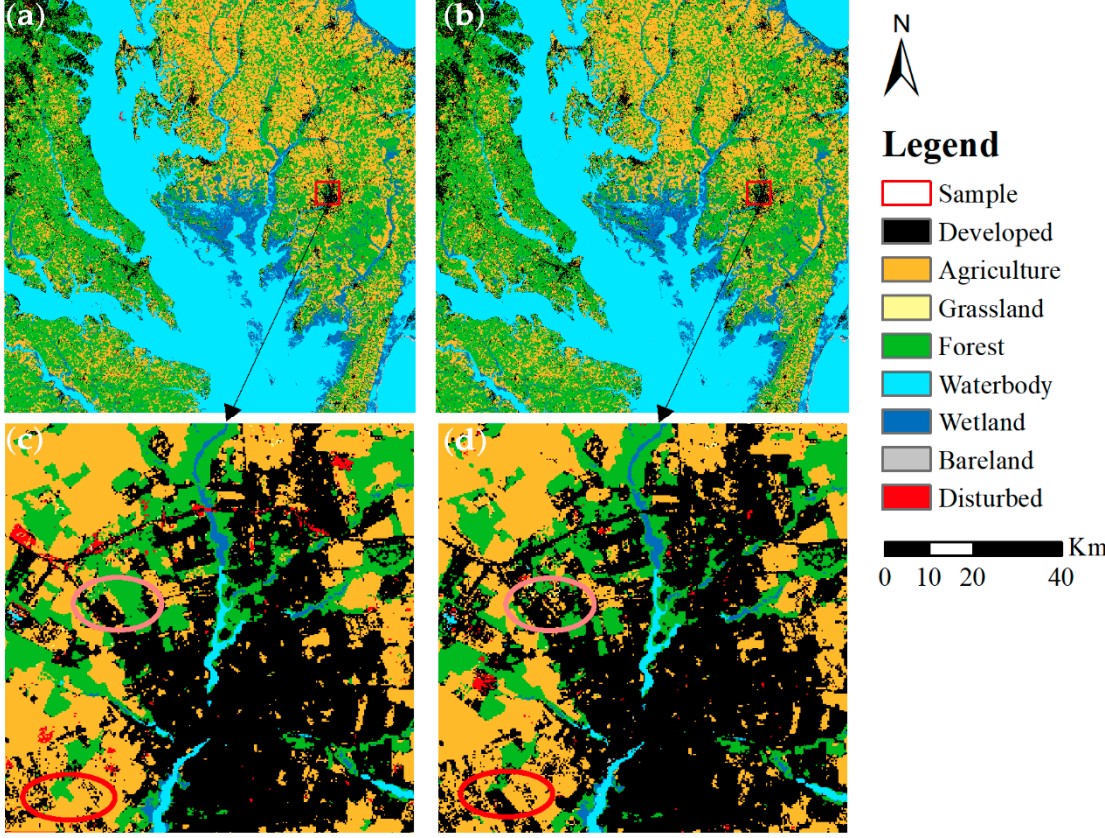

**Figure 3.** Comparison of LULC for LCMAP in 2001 and 2011. (**a**) shows land cover in 2001, with (**b**) land cover in 2011. (**c**) and (**d**) are zoomed-in figures of interest areas in (**a**) and (**b**), respectively. Ovals in (**c**) and (**d**) show the process of urbanization (pink ovals show conversion from forest and red ovals the conversion from forest and agriculture to urban land use).

**Table 1.** Main area (km$^2$) LULC matrix comparing LCMAP (2001–2011).

| From Class 2001 | To Class 2011 | | | | | | |
| --- | --- | --- | --- | --- | --- | --- | --- |
| | Developed | Agriculture | Grassland | Forest | Wetland | Disturbed | Sum Area (2001) |
| Developed | 1269 | 20 | 1 | 20 | 3 | 4 | 1316 |
| Agriculture | 122 | 5509 | 5 | 31 | 1 | 22 | 5689 |
| Grassland | 3 | 4 | 11 | 3 | 0 | 0 | 22 |
| Forest | 168 | 34 | 5 | 6193 | 10 | 30 | 6442 |
| Wetland | 25 | 2 | 1 | 50 | 1643 | 8 | 1728 |
| Disturbed | 21 | 63 | 1 | 55 | 6 | 3 | 149 |
| Sum Area (2011) | 1609 | 5633 | 23 | 6351 | 1663 | 67 | 15,346 |

### 3.2. Carbon Stock Trends of CBW

Land-cover types, including agriculture, developed, grassland, forest, wetland and disturbed (only in LCMAP), were considered to access the carbon stock in LCMAP. TEC is the sum of carbon stored in living biomass, standing and downed dead wood biomass, raw litter, litter, and soil (of 2-m depth). Units of the carbon (C) pool were $10^6$ kg C km$^{-2}$ (kg C m$^{-2}$). For the TEC of 2001, there was 344 Tg C in terrestrial carbon storage, of which 42% was in forest, 26% in wetland and 26% in agriculture. By 2011, TEC increased to 357 Tg C, with the largest increases in litter (5 Tg C), living biomass (4 Tg C) and soil (3 Tg). Regarding land cover, the largest increases were forest (8 Tg), with the largest decrease in agriculture (2 Tg) and wetland (1 Tg) (Figure 4).

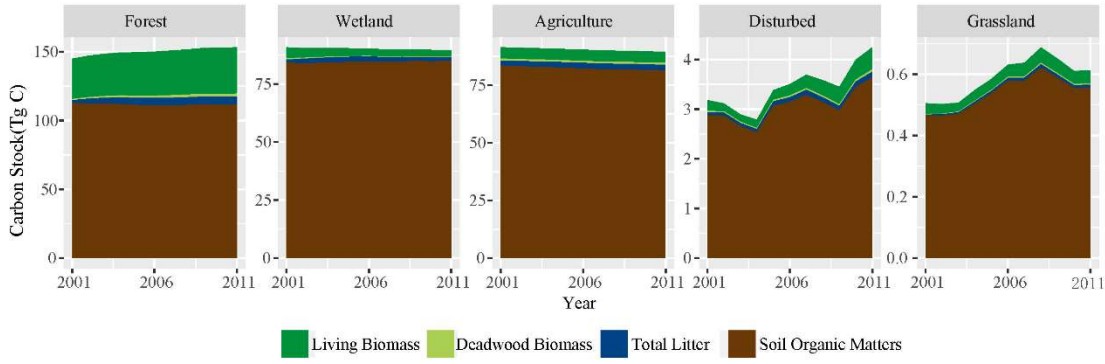

**Figure 4.** Carbon storage overtime organized by LULC class.

### 3.3. Carbon Stock and Flux Caused by LULCC

Our case study provided a sample of carbon stock types (Figures 5 and 6) and carbon emissions (Figure 7) that could be simulated using LUCAS. The results showed that the TEC for each land cover was relatively stable in the reference scenario, with static land cover in 2001. However, both TEC and emissions in the LULCC scenario for forest land, wetland, agriculture land, disturbed land, and grassland varied from those in the reference scenario. The TEC of all land covers decreased at an annual rate of 0.04 Tg C·yr$^{-1}$. The greatest carbon loss (0.22 Tg C·yr$^{-1}$) was due to forest area decrease, followed by wetland (0.24 Tg C·yr$^{-1}$) and agriculture (0.18 Tg C·yr$^{-1}$) (Figure 6). The major carbon emissions resulted from agricultural expansion (0.046 Tg C·yr$^{-1}$) and urbanization (0.224 Tg C·yr$^{-1}$) (Figure 7). The largest carbon emission was produced by the conversion from forest to developed land (0.157 Tg C·yr$^{-1}$), followed by urbanization from agriculture (0.0497 Tg C·yr$^{-1}$).

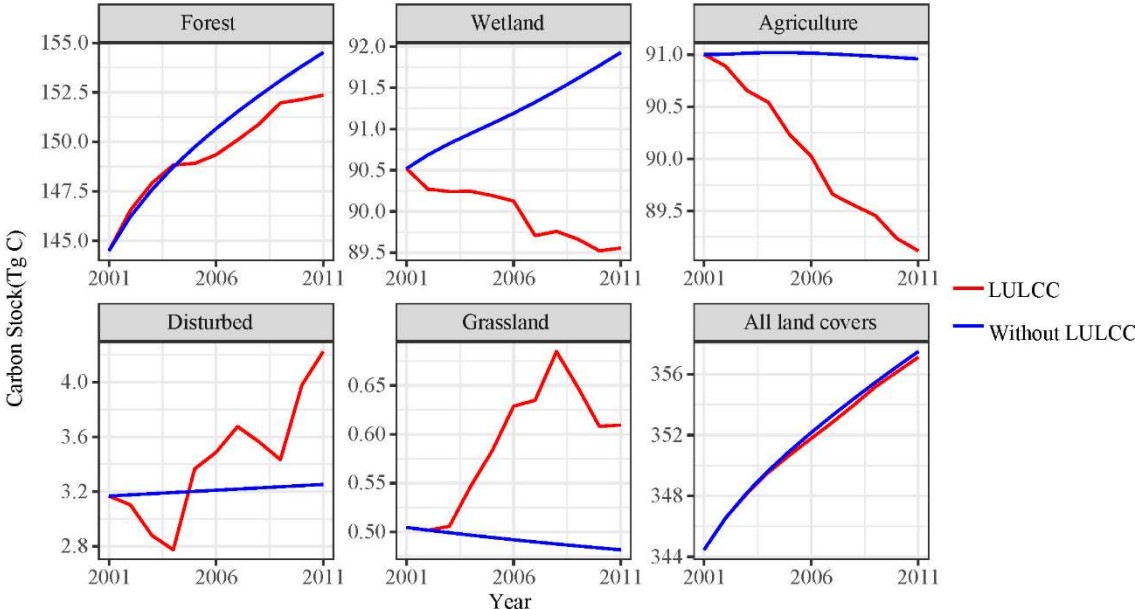

**Figure 5.** Carbon trends simulated under two land-cover change scenarios. Scenario LULCC entails simulating carbon stocks based on the annual LULCC data from LCMAP. Scenario without LULCC means simulating carbon stock based on land covers in 2001 from LCMAP. LULCC; land-use and land-cover change.

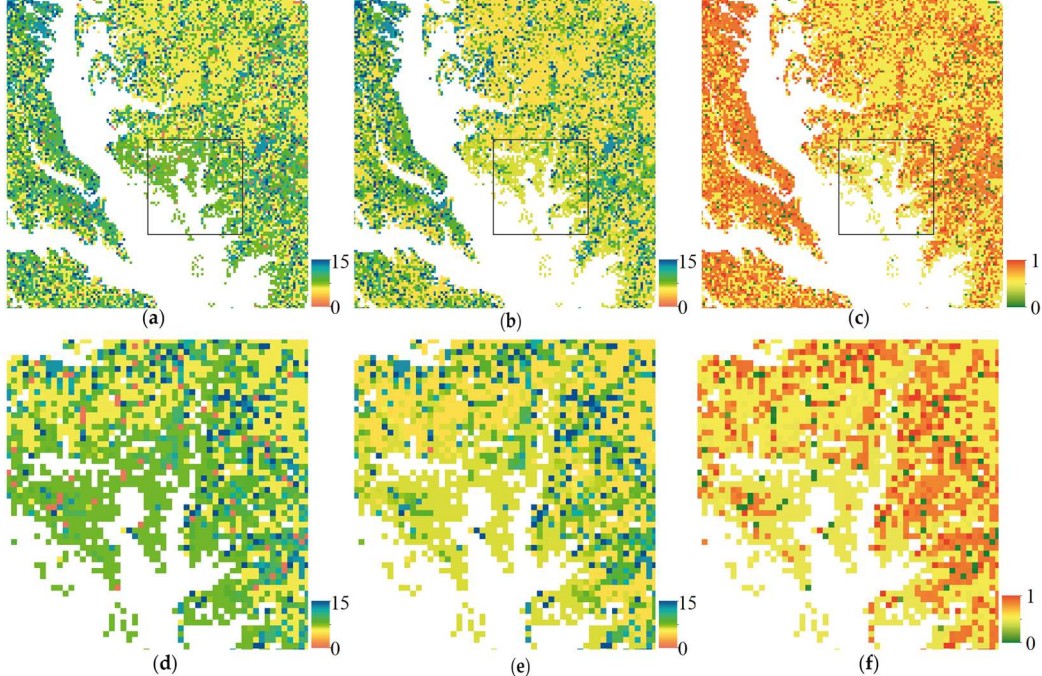

**Figure 6.** Comparison of ratio of total ecosystem carbon storage (TEC) in different scenarios. (**a**) Ratio of TEC in scenario LULCC of 2011 and initial TEC of 2001; (**b**) ratio of TEC in scenario without LULCC of 2011 and initial TEC of 2001; (**c**) ratio of TEC in scenario LULCC of 2011 and TEC in scenario of LULCC in 2011; (**d**), (**e**) and (**f**) are zoomed-in figures of interest areas in black squares, respectively, of (**a**), (**b**), and (**c**).

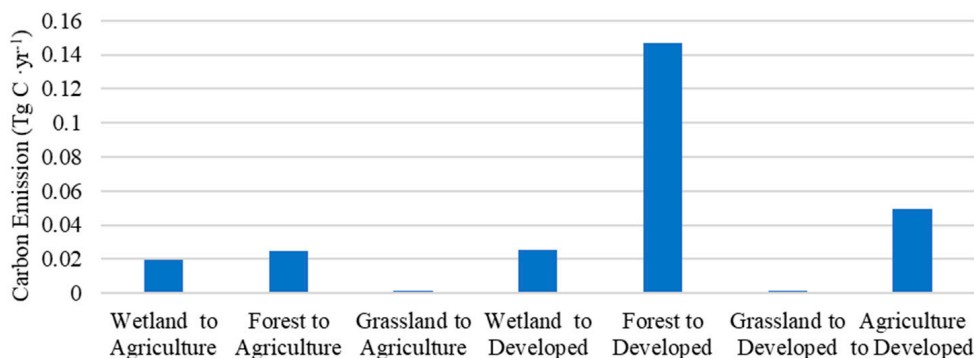

**Figure 7.** Annual carbon emissions due to land use change.

### 3.4. Impacts of Time Density of LULC on Land-Cover Change and Carbon Flux

Calculation of the land transition rates was dependent upon the source data type and time interval. Major land transitions were the conversion between forest and disturbed land in LCMAP. With the density of LULC input, more transitions between forest and disturbed land were recognized (Figure 8). Other land conversions fluctuated with changes in time interval. Regarding conversions between forest and disturbed land from LCMAP, the changed areas varied by annual (283 km$^2$), 2–3-year (117 km$^2$), 5-year (75 km$^2$), and 10-year (55 km$^2$) intervals (Figure 9).

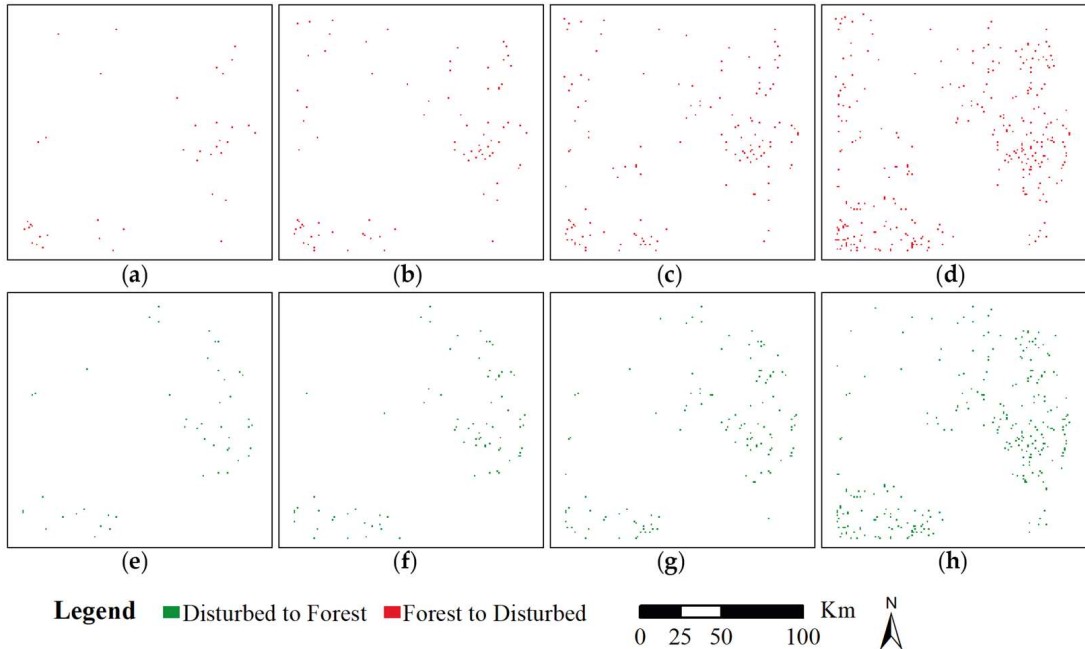

**Figure 8.** Geographic comparison of land transition area resulting from various time intervals. (**a**), (**b**), (**c**), and (**d**) are the sum of forest increase resulting from respective scenarios of 10 yrs, 5 yrs, 2 yrs, and annual. (**e**), (**f**), (**g**) and (**h**) are the sum of forest decrease resulting from respective scenarios of 10 yrs, 5 yrs, 2 yrs, and annual.

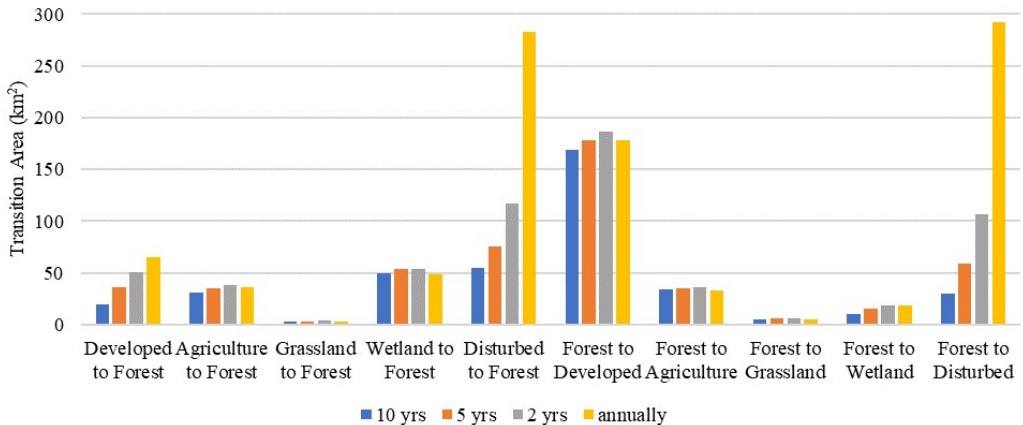

**Figure 9.** Comparison of land transition area resulting from various time intervals.

Overall, the greatest carbon emissions (> 0.8 Tg) during 2001 to 2011 occurred in forest to developed transitions, whereas the remaining land transitions had lesser emissions (0.4 Tg; Figure 10). For the effect of time frequency of land-cover maps on carbon emissions, the results show that the transitions of forest to developed land and forest to disturbed land had increased. The transitions of forest land to agricultural land fluctuated.

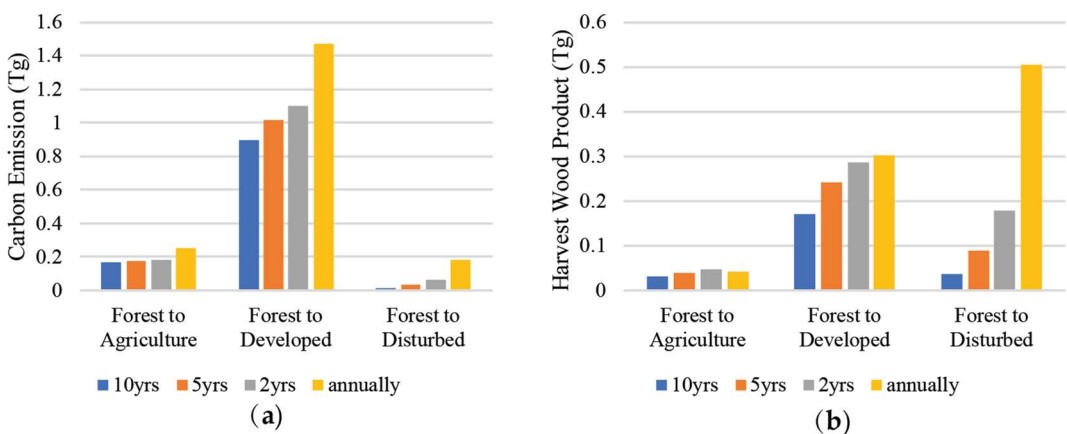

**Figure 10.** Comparison of carbon emissions (**a**) and harvest wood products (**b**) resulting from LULCC with time density and data source.

The HWP from urbanization and conversion from forest was much greater than that from other LCMAP-induced land changes. The effect of time frequency on HWP shows that the transitions between forest and wetland and forest to grassland increased with the time density of LULCC, as well as the transitions of forest to developed land and forest to disturbed land in LCMAP. The other transitions, such as forest and agriculture, fluctuated.

## 4. Discussion

The CBW has undergone great changes due to human activities and population growth. In this study, we found that the total ecosystem carbon of the CBW increased (13 Tg C) from 2001 to 2011, largely attributable to the carbon sink of the forest ecosystem. Our results concur with other studies showing that LULCC affected the overall biomass carbon stock in the historical period [30], owing to substantial natural vegetation conversion to cropland by human activities [31]. It has been estimated that 145 Pg C carbon was lost from the terrestrial ecosystem as a result of LULCC worldwide since 1850 [32]. LULCC and disturbance caused a loss of $1.11 \pm 0.35$ Pg C yr$^{-1}$ annually [32]. Lister et al. [33] indicated that the Chesapeake Bay Basin had a net loss of 2% of forest land since

the 1980s. Claggett et al. [25] predicted that, to a large extent, forests in the Gulf region of CBW will be severely disrupted in the coming decades because of development pressure. Our study shows that urbanization of agricultural and forest lands is the main source of carbon emissions in total ecosystem carbon, which is similar to another study of the Chesapeake Bay [24]. Unlike natural land transitions without abrupt removal of aboveground biomass (such as from forest to wetland or grassland), anthropogenic land transitions, including agricultural expansion and urbanization, would remove carbon from the ecosystem, which has been considered the main carbon emission. In our study, by comparing the LULCC and no land-cover change scenarios, results showed that total carbon had decreased by 6.4 Tg from 2001 to 2011. This is mainly attributable to urbanization and agricultural expansion, consistent with other studies [24,30,31,34].

There was 123 km$^2$ of agricultural land converted to developed land from 2001 to 2011; this is very similar to that of NLCD (104 km$^2$). However, because the national wetland inventory is derived using different methods and thresholds, the consistency of LCMAP is less than that of wetland mapping results obtained by NLCD [10]. Thus, some pixels near the waterbody were classified as wetland in LCMAP but forest in NLCD. Rhode Island's forests are overestimated by almost 8%, and most misclassifications occur in the NLCD forest category, which is actually developed for agricultural land [35]. Because of the impacts of LULC on environmental processes, such as carbon dynamics and the hydrologic cycle, high-frequency land-cover mapping products are important for reducing model uncertainties [36–38]. The widely used NLCD land-cover product is available at a five-year interval since 1992, which helps quantify state and land change between two observation times. However, some land-cover changes or management activities cannot be detected because the NLCD data are not temporally dense enough. The LCMAP product provides comprehensive annual and sub-annual land-cover change information that enhances our understanding of recent historical land-change processes [11,39]. Therefore, high temporally dense LULC products can help reduce uncertainties in quantifying land and carbon changes.

One of the ability of LUCAS is to estimate the carbon uncertainty resulting from the uncertainty of land-cover map, i.e., the probability of land transitions between two observation times. Since we are using the new annual LCMAP product, we did not have any transition probabilities used. The overall land change uncertainty is inherited from the original LAMAP data [11,12]. LUCAS has been used to estimate the impact of LULC change on carbon stock and flux in accordance with other studies for the state of California from 1975 to 2010 [22,40–42]. The IBIS model can track the impact of carbon through a common set of ecosystem pools with accuracy > 0.9 in simulating NPP and living biomass [43]. Comparing to similar studies, our estimates of NPP (560 to 590 g·m$^{-2}$·yr$^{-1}$) is close to the results of the York River [44], but smaller than other studies [45,46] with ranges from 1000 to 1500 g·m$^{-2}$·yr$^{-1}$. There are other methods, including model and remote sensing, to estimate NPP. Since the uncertainty of NPP is important for estimating ecosystem carbon balance, we believe improving NPP inputs will allow better carbon accounting. Currently, we only used the NPP derived from the IBIS model. In the future, we plan to use LUCAS to quantify the uncertainty of carbon balance with multiple NPP inputs derived from remote sensing (e.g., MODIS product MOD17A3 [47]) and other process models (e.g., DAYCENT [48] and CLM [49]).

High-resolution land-cover product can quantify more landscape details than Landsat based land cove product. However, its higher cost of data acquisition and processing limits its production over large area and in a long-term manner. Thus, annual high-resolution land-cover products are hard to obtain, let alone using them to assess carbon emission caused by land-cover changes. Currently, most land-cover products including NLCD, LCMAP, Globcover30 [50], Globcover2005 [51], GLC2000 [52] are in medium or coarse resolution. These products have suitable spatial and temporal resolutions for quantifying carbon emissions. The current results can be used to improve monitoring and targeting and assist in adaptive management practices designed to change behavior, especially where maps show the greatest changes. These high-density LULC data allow more detailed land-change-related carbon accounting than less-dense LULC data. Land-use changes and associated carbon change have

not been previously evaluated at this level of spatial detail. Approaches identical or similar to ours can be used in other areas, especially where land-cover change is frequent.

## 5. Conclusions

We used high-density LULC data to estimate the effects of land-cover transitions on the terrestrial ecosystem carbon balance of the southern Chesapeake Bay region from 2001 to 2011. Combined analysis of land-cover and carbon changes have not been previously completed at this level of spatial detail. Changes in ecosystem carbon in the study area were mainly driven by land transitions, especially the shrinkage of forest and agriculture. We found that 90 km$^2$ of forest land, 57 km$^2$ of agricultural land, and 65 km$^2$ of wetland were lost to development from 2001 to 2011. By tracking land-use changes and consequent effects on carbon stocks and flows relative to the no land-cover change scenario, we found that LULCC had the greatest impacts on the carbon stock in forest (reduction of 0.35 Tg C·yr$^{-1}$). LCMAP provides 30-m resolution annual land-cover transitions. These unique products allow more detailed land change-related carbon accounting than in previous carbon studies. Our approach can thus be used in areas where land-cover change is frequent.

The LUCAS model is an efficient tool for spatial and temporal land-change and carbon simulation. LUCAS can quantify a variety of model and data uncertainties, including the relative impact of data uncertainties on carbon flux. Our modeling study is the first of its kind to focus on the carbon cycle as well as land-cover change using LCMAP data and the LUCAS model. The LUCAS approach can be used to investigate a broad range of future carbon scenarios, such as assessing future climate mitigation strategies by establishing an empirical baseline.

**Author Contributions:** Conceptualization, J.D., J.L. and Z.Z.; Data curation, J.L. and Z.Z.; Formal analysis, J.D. and J.L.; Funding acquisition, M.L., Z.Z. and J.L.; Methodology, J.L. and B.M.S.; Software, J.D., J.L. and B.M.S.; Writing—original draft, J.D.; Writing—review & editing, J.L., Z.Z., and M.L. All authors have read and agreed to the published version of the manuscript.

**Funding:** This work was jointly funded by the following grants: The National Natural Science Foundation of China [31670552,31971577], the DOD ESTCP Program (RC_201703), the doctorate Fellowship Foundation of Nanjing Forestry University [2016], and the PAPD (Priority Academic Program Development) of Jiangsu Provincial Universities [2017].

**Acknowledgments:** The authors would like to acknowledge the Land Change Monitoring, Assessment, and Projection (LCMAP) for providing the initial test data. This work was implemented when the corresponding author acted as an awardee of the 2017 Qinglan Project sponsored by the Jiangsu province. Special thanks to APEX Resource Management Solution Ltd. for providing SyncroSim software and ST-Sim State-and-Transition Simulation Model.

**Conflicts of Interest:** The authors declare no conflict of interest.

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
