# Peer review of "Substantially Greater Carbon Emissions Estimated Based on Annual Land-Use Transition Data"

_remotesensing, doi:10.3390/rs12071126_

Round 1

Reviewer 1 Report

In general, the article is well written, the subject matter is within the scope of Remote Sensing journal. However, a few elements of the article should be improved.

Taking into account the content of the research, the use of the word CO2 in the title seems to be unjustified. Instead, it would be more appropriate to use 'carbon emission' , 'carbon stock' or 'carbon fluxes' as a result of changes in annual land use transition data.

It is not clear enough what the novelty in the presented studies is. The assessment of land use changes in time, based on data from Landsat satellites, is a common undertaking. Using the LUCAS model to estimate changes in carbon stock or emissions would be such an element. However, I think it is necessary to include precise characteristics of the model and its current applications. The first reference to the LUCAS model concerns research on landscape dynamics [18]. However, in this article there are no references to the LUCAS model. The Works Cited should be reviewed and supplemented in this respect.

On what basis was it assumed that "cells with a negative forest age were assigned an initial age of 50 years"?

How to explain the step (about 30%) change in 'soil organic matters' in the total 'carbon storage' value for the 'disturbed' class around 2003-2004 (figure 4)? Should this category be included in this figure?

Line 320: LCMAP provides 30-resolution. Shouldn't this be 30 – m resolution?

I think the discussion lacks an assessment of the accuracy of the method and the results of carbon emissions, which, however, are based on the inaccurate 30 m resolution of Landsat imaging.

Reviewer 2 Report

In general terms, the research is well developed. The addressed topic is focal and of great interest for Remote Sensing readers. It is in line with some of its subject areas such as “change detection”, “remote sensing applications”. Even with other issues that, although not explicitly stated by Remote Sensing, have a general interest for scientists and citizens: the LULC changes evaluation, taking into account different scenarios, in the context of Global Change and the Global Carbon Program.

The manuscript describes the LULC changes, in the first decade of our century, in the United States, Chesapeake Bay, a rapidly changing region affected by human activities. It uses the cartographic series LCMAP, from the U.S. Geological Survey it takes into account two scenarios. On the other hand, it analyzes the effects of LULC changes on carbon stocks and flows and their trends. To do this, it uses different models and simulators (LUCAS and IBIS) and takes into account three scenarios. Analyze the results of the main transitions between LULC and its effects on the carbon cycle, using different time series: annual, bi-annual, five-year, ten-year.

In my opinion, the originality and the main value of this paper rely on the relationship between LULC changes and their effects on carbon stocks and flows. The results provide very relevant environmental information for the managers of this large river basin and for the planners of urban areas and their surroundings. As the authors say in the conclusions, the results can guide future strategies focused on climate mitigation in the region.

The methods used are correct and designed to solve a specific environmental challenge. The article is organized around a logical and traditional structure. It is well documented and provides appropriate references.

However, a minor revision is advisable to improve the article. Next, I propose a series of suggestions as a guide for the review:

  • Explain the acronym DGVM (line 77). I guess it means Dynamic Global Vegetation Model.
  • Perhaps, the authors could reflect, in the "Discussion" section, on the opportunity to use other alternative products to calculate the NPP (e.g. MODIS product MOD17A3).
  • The paragraph of lines 193-202 repeats, in text mode, the data in table 1. I suggest the authors detail here the processes behind them and explain these transitions.
  • In Figure 3, it is difficult to observe the differences between a) and b). I recommend adding some windows, as a zoom, in which the authors illustrate in detail some relevant transitions.
  • Similarly, I propose to enlarge Figure 6. Below it, I suggest adding a zoom in a significant area where the differences between a-b (on the left), a-c (in the center), b-c (on the right) are shown.
  • In Figure 7, I suggest moving the "Wetland to Developed" bar to the left of the "Forest to Developed" bar. Thus, the same sequence will be repeated: "Wetland to ..." "Forest to ..." "Grassland to ..."
  • In the titles of figures 8 and 9, I think it would be advisable to add "from/to forest" for easy reading.
  • At the bottom of Figure 9, I think the authors should add the legend that says "10 yr, 5 yr, 2-3 year, annual".
  • Considering all the time series, in my opinion, the most relevant thing in Figure 8 is that the greatest transitions occur from forest to developed, in a consistent way. I suggest that the authors explain this process from a geographical perspective.
  • On line 255, I understand that where it says ".... and conversion form forest" it should say ".... and conversion from forest"
  • Should not the authors indicate Hollister et al. 2004, as a numerical reference? (Line 271). In addition, it does not appear in the reference list.
  • The sentence of lines 301-304 is unnecessary in the "Discussion" section. Repeats on line 311 and following.
  • I suggest replacing "30-resolution" with "30-m resolution" (lines 274 and 320).
  • I recommend replacing "chesapeake bay watershed" with "Chesapeake bay watershed" (line 422).
